# Cost savings in male circumcision post-operative care using two-way text-based follow-up in rural and urban South Africa

**Yanfang Su**[1]☉, **Rachel Mukora**[2]☉, **Felex Ndebele**[2], **Jacqueline Pienaar**[2,3], **Calsile Khumalo**[2], **Xinpeng Xu**[4], **Hannock Tweya**[1,5], **Maria Sardini**[3], **Sarah Day**[3,6], **Kenneth Sherr**[1], **Geoffrey Setswe**[2,7], **Caryl Feldacker**[1,5]*

1 Department of Global Health, University of Washington, Seattle, WA, United States of America, 2 The Aurum Institute, Johannesburg, South Africa, 3 Centre for HIV-AIDS Prevention Studies (CHAPS), Johannesburg, South Africa, 4 School of Public Health, Nanjing Medical University, Nanjing, China, 5 International Training and Education Center for Health (I-TECH), Seattle, WA, United States of America, 6 School of Public Health, Faculty of Health Sciences, University of Cape Town, Cape Town, South Africa, 7 Department of Health Studies, University of South Africa (UNISA), Pretoria, South Africa

☉ These authors contributed equally to this work.
* cfeld@uw.edu

**Data Availability Statement:** All relevant data are within the manuscript and its Supporting Information files.

## Abstract

### Introduction

Voluntary medical male circumcision (VMMC) clients are required to attend multiple post-operative follow-up visits in South Africa. However, with demonstrated VMMC safety, stretched clinic staff in SA may conduct more than 400,000 unnecessary reviews for males without complications, annually. Embedded into a randomized controlled trial (RCT) to test safety of two-way, text-based (2wT) follow-up as compared to routine in-person visits among adult clients, the objective of this study was to compare 2wT and routine post-VMMC care costs in rural and urban South African settings.

### Methods

Activity-based costing (ABC) estimated the costs of post-VMMC care, including counselling, follow-ups, and tracing in $US dollars. Transportation for VMMC and follow-up was provided for rural clients in outreach settings but not for urban clients in static sites. Data were collected from National Department of Health VMMC forms, RCT databases, and time-and-motion surveys. Sensitivity analysis presents different follow-up scenarios. We hypothesized that 2wT would save per-client costs overall, with higher savings in rural settings.

### Results

VMMC program costs were estimated from 1,084 RCT clients: 537 in routine care and 547 in 2wT. On average, 2wT saved $3.56 per client as compared to routine care. By location, 2wT saved $7.73 per rural client and increased urban costs by $0.59 per client. 2wT would save $2.16 and $7.02 in follow-up program costs if men attended one or two post-VMMC visits, respectively.

**Funding:** Research reported in this publication was supported by the National Institute of Nursing Research (NINR) of the National Institutes of Health under award number 5R01NR019229, "Expanding and Scaling Two-way Texting to Reduce Unnecessary Follow-Up and Improve Adverse Event Identification Among Voluntary Medical Male Circumcision Clients in the Republic of South Africa." There was no additional external funding received for this study. The content is solely the responsibility of the authors and does not necessarily represent the official views of the National Institutes of Health. The funders had no role in study design, data collection and analysis, decision to publish, or preparation of the manuscript.

**Competing interests:** The authors have declared that no competing interests exist.

**Abbreviations:** 2wT, Two-way texting; AE(s), adverse event(s); CHAPS, Centre For HIV-AIDS Prevention Studies; VMMC, male circumcision; NDoH, National Department of Health; PEPFAR, President's Emergency Plan for AIDS Relief; SA, South Africa; WHO, World Health Organization.

## Conclusion

Quality 2wT follow-up care reduces overall post-VMMC care costs by supporting most men to heal at home while triaging clients with potential complications to timely, in-person care. 2wT saves more in rural areas where 2wT offsets transportation costs. Minimal additional 2wT costs in urban areas reflect high care quality and client engagement, a worthy investment for improved VMMC service delivery. 2wT scale-up in South Africa could significantly reduce overall VMMC costs while maintaining service quality.

## Introduction

Voluntary medical male circumcision (VMMC) is one of the most successful biomedical prevention strategies to reduce HIV transmission risk [1, 2]. VMMC is also safe with reported rates of combined moderate and severe adverse events (AEs) in large VMMC programs operating at scale in Southern Africa below 2% [3–7], meaning that 98% of VMMC clients likely heal without any incident. The Republic of South Africa SA prioritizes VMMC as part of its comprehensive HIV prevention, treatment and care plan [8], completing over 400,000 VMMCs per year for the last 10 years [9]. National South Africa guidelines require all VMMC clients to attend two post-operative follow-up visits within 14 days after the procedure to ensure timely identification and treatment for AEs. However, these visits are costly to both the clients and the providers. To comply with guideline requirements, attendance at post-operative reviews may be overreported, potentially inflating review rates while reducing the quality of care [10, 11]. A cost-effective approach is needed to reduce unnecessary visits while maintaining care quality.

Mobile health (mHealth) is used to describe the practice of healthcare and public health supported by mobile communication devices such as cellphones. mHealth interventions are recognized as more cost-effective compared to conventional interventions and may help address persistent challenges of healthcare worker (HCW) shortages and client difficulty in accessing healthcare facilities [12, 13], including in the VMMC context [14]. From 2009 to 2019, thousands of mHealth interventions were documented and launched globally, the vast majority in resource-constrained settings, suggesting robust approval of, and desire for, digital technologies in healthcare [15–17]. When implemented optimally, mHealth interventions may help reduce healthcare costs by improving the provision of health education, identifying health concerns early when illness may be less severe, reducing the duration of therapy, and minimizing transport costs [18].

In 2018, the International Training and Education Center for Health (I-TECH) in the Department of Global Health at the University of Washington (UW) and technology partner, Medic, conducted a randomized control trial (RCT) among VMMC clients in routine VMMC clinics in Zimbabwe using a two-way texting (2wT) telehealth approach between clients and nurses. 2wT is an mHealth platform for conversational messaging between VMMC clinicians and clients, providing short messaging service (SMS) for post-operative telehealth care in lieu of scheduled, in-person visits. In Zimbabwe, and likely in other VMMC programs operating at scale, in-person, post-operative visits may be burdensome for patients and providers, leading to poor attendance [10, 11]. 2wT improved the quality of post-operative care while reducing HCW visit workload [19]. The 2wT approach for VMMC follow-up in Zimbabwe reduced VMMC program costs while improving client care engagement, with a net savings of $2.10

using 2wT over routine follow-up visits [14]. 2wT was also found to be highly usable and acceptable among clients and providers [20], reaching >31,000 VMMC clients in Zimbabwe by 2023 [21].

South Africa, with reported mobile-cellular subscriptions of 161.8 per 100 population in 2020 [22] is a good candidate for exploring cost-effective mobile phone innovations, like 2wT for VMMC. To test the 2wT approach in South Africa, I-TECH conducted a pragmatic RCT to test 2wT in urban and rural settings in South Africa in partnership with the Aurum Institute, the Centre for HIV-AIDS Prevention Studies (CHAPS) and Medic. Previously, we disseminated results from clinical and usability outcomes of the 2wT RCT in South Africa where 2wT improved the quality of post-operative care in both rural and urban settings, increasing the ascertainment of AEs, engaging males in follow-up care, and showing high acceptability for clients and providers [23, 24]. In growing recognition of the importance of costing research to inform HIV prevention interventions globally and in South Africa, we embedded a costing study within the South Africa 2wT RCT.

Costing of HIV prevention interventions is gaining recognition for its importance in informing policy and scale-up. In recent reviews of costing research on HIV prevention interventions in sub-Saharan Africa, 38 out of 159 studies (24%) were conducted in South Africa [25, 26]. Although there is clear evidence for VMMC as a cost-effective HIV reduction intervention [27, 28], to our knowledge, there is no costing research on innovations to improve efficiency of VMMC care delivery in South Africa. This costing study is nested in the clinical 2wT RCT [23]. The objective of this embedded study was to estimate the costs associated with 2wT mHealth follow-up as compared to routine in-person VMMC follow-up in rural and urban South Africa. While some clients may pay for transport for VMMC services in urban areas where services are typically more convenient, in rural areas, it is common that VMMC programs provide transportation for both the VMMC procedure and follow-up visits to increase VMMC uptake [29, 30]. The VMMC program in SA operates in predominantly rural areas [23]. As with the previous Zimbabwe study [14], the payer perspective is appropriate for this analysis as it is largely the VMMC program (the payer) that bears these follow-up costs and stands to gain the most from the 2wT approach to reduce unnecessary visits. As VMMC costs are largely provided by the global donor community, including the US President's Emergency Plan for AIDS Relief (PEPFAR), and to mirror the approach taken to assess costs benefits of 2wT for VMMC in Zimbabwe, we assessed costs from the payer perspective. This approach is also in line with recent evidence suggesting continuing donor support for VMMC costs for at least 5 additional years given the cost-effectiveness of the overall VMMC intervention for HIV prevention [31].

## Materials and methods

We applied the activity-based costing (ABC) approach to estimate the costs associated with 2wT intervention. We focused on estimating the marginal cost of follow-up for an additional VMMC client using the 2wT or routine follow-up approach.

### Comparators

**Routine care.** The VMMC implementing partner, Centre For HIV-AIDS Prevention Studies (CHAPS), followed all National Department of Health (NDoH) protocols [32] for routine VMMC services, post-operative wound care counselling, in-person follow-up visits on post-surgery days 2 and 7, and tracing. Routine VMMC data collection defines the post-operative day 2 visit window from day 1 to day 4 and the day 7 visit window from day 5 to day 10. Typically, in urban CHAPS VMMC services, clients pay for their own transportation costs to

**Table 1. Activities undertaken in the 2wT study.**

| Activities | Routine | 2wT |
|---|---|---|
| Day of male circumcision | | |
| Adverse event (AEs) identification counselling | Yes | Yes |
| Counselling on SMS interactions and photos of potential AEs named in daily message | No | Yes |
| Post-operative follow-ups | | |
| Required day 2 and day 7 in-person visits | Yes | No |
| In-person reviews for potential AEs on any day | Yes | Yes |
| Day 1–13 SMS | No | Yes |
| AE treatment | Yes | Yes |
| Tracing via phone calls and home visits for lost to follow up (LTFU) | Yes | Yes |

attend the VMMC procedure visit and any follow-up visits, routine or potential AE related. In rural areas, the CHAPS team typically provides transportation both for the procedure visit and any follow-up, routine or potential AE related. Exceptions to transportation generalities in either setting do occur. NDoH policy requires participant tracing by phone calls and/or home visits, for those who do not attend post-operative reviews on days 2 and 7 in routine care (Table 1).

**2wT intervention and specific 2wT RCT procedures.** The 2wT approach for VMMC follow-up was described previously [19, 21, 23]. 2wT is a hybrid mHealth model integrating both automated SMS and individualized mHealth messaging between clients and a routine VMMC nurse. On day 0, 2wT clients were enrolled into a custom 2wT software application built using the open-source Community Health Toolkit (CHT). Before leaving the VMMC clinic, 2wT clients received enhanced post-operative counselling based on global VMMC guidelines [33] using the study-specific 2wT flipbook with additional wound care guidance, photos of AE warning signs, and instructions on how to respond to the daily SMS (Table 1). 2wT specific counseling helped 2wT clients understand the automated daily message that asked about the five common complications (i.e., bleeding, swelling, pus, redness, and wound opening), preparing men to take ownership over their wound care. 2wT participants received an automated daily text on days 1–13 in either English, Setswana, or isiZulu, and could respond in any language. Daily automated text messages read, "are you experiencing any bleeding, swelling, pus, pain, redness, wound opening, or other concern? Enter 1 = Yes; 0 (Zero) = No / I'm Ok, and press send." Receiving and sending text messages was free for the participants. The SMS prompt encouraged the clients to reply with binary answers (0 for No, there is not a problem and 1 for Yes, there is a problem). Nurses communicated further via SMS with clients who reported any concerns and also made phone calls if needed. 2wT clients were not required to attend any clinic visits but could attend the clinic at any time if they needed. For 2wT men with a potential AE or with the need for further reassurance, the nurse triaged the participants for clinical visits the following day or earlier if an emergency was suspected. If 2wT clients did not respond to text messages by day 8, phone tracing was activated by the clinical VMMC team. All 2wT participants, in both arms, were asked to return for a study-specific visit on post-operative day 14 at which point a US$7 cell phone credit was given to all participants to compensate for time and travel. Participants were not traced via home visit for missing the study-specific day 14 visit. The day 14 visit window included visits from day 13 until day 21. Participants were declared off-study after day 21 or after completion of the day 14 visit.

An *Enrolled* Nurse cadre level led the 2wT interaction and patient follow-up activities at the rural site while a *Professional* Nurse cadre level led the follow-up activities at the urban site with additional supervisory duties across study sites.

## Data collection and assumptions

While exceptions do occur, in general, it is most common that VMMC clients self-paid transportation costs in urban sites and that the program paid for client transportation costs in rural areas. These transportation commonalities are reflected in the payer perspective, representing CHAPS costs as part of their overall VMMC program costs funded by the major VMMC donor, PEPFAR. This is the most prevalent model of VMMC funding and was assumed for the current approach.

For the costing study, data were collected from three main sources: 1) RCT databases of routine VMMC data including client visits, AEs, and follow-up tracing; 2) the 2wT database; and 3) time-and-motion surveys designed for the costing component of the study. The first source, the RCT databases, included de-identified routine VMMC data on the number of visits attended, the number/type/severity of AEs, and the number of traced clients, by tracing method (phone and/or home visit). Visit costs associated with the study day 14 visit were excluded since it was specific to the RCT study and not considered routine VMMC follow-up care. The 2wT database contained enrolment data including transportation costs to the clinic. The 2wT database also included counts of inbound and outbound SMS. RCT-specific activities (e.g., additional consenting, RCT monitoring forms) were also not included in time-and-motion data.

Based on Time and Motion Form (S1 File), data collection tools were created using both Epicollect5 and Microsoft Excel and were piloted before data collection. Time-and-motion studies consisted of five days of direct observations at both rural and urban sites. We collected data about distance travelled using car odometer readings as well as time spent in travelling and conducting client reviews in the rural district of Bojanala, South Africa, between 29th November 2021 and 12th January 2022. Nurses took notes to specify how time was spent in each activity, providing additional qualitative data inputs for costing analysis. We collected reception time and nurse review time at the urban district of Ekurhuleni, South Africa, between 12th January 2022 and 24th January 2022.

It is assumed that (1) all fixed costs related to post-operative follow-ups were in place, and hence we focus on analysing marginal costs in serving an additional VMMC client in the system; (2) that full-time employees worked a 40-hour workweek or 160 hours per month; (3) that nurse counselling time was 5 minutes in routine counselling and 10 minutes for 2wT counselling; (4) that the nurse spent one minute per SMS response on average, and (5) that the average phone call time with a VMMC client was 5 minutes.

**Hypotheses.** We tested the following two hypotheses in our costing study:

Hypothesis 1: 2wT reduces post-VMMC care costs by triaging only those in need of in-person review to care, allowing most men to heal safely at home without in-person follow-up visits.

Hypothesis 2: 2wT saves more costs in rural VMMC program settings than in urban program settings.

## Data analysis: Activity-based costing

Using an ABC approach, we estimated the costs in the post-VMMC care continuum from the perspective of the payer–the VMMC program with donor support. There were four activity categories in costing: counselling on the day of male circumcision, SMS follow-ups, physical follow-up visits, and tracing. All activity-based costs were estimated for 2wT care and routine care to test the hypotheses of cost savings. We applied ranges of ±50% uncertainty interval for parameters presented in Table 2 [14]. Input data were analysed using STATA version 14.0 and costing was conducted in Microsoft Excel. Editable costing tool (S1 Table) presented our input

**Table 2. Summary statistics by activity category in rural and urban South Africa.**

| Parameters | Sum | Average | Rural | Urban | Source |
|---|---|---|---|---|---|
| | N (Uncertainty Interval) | N (UI) | N (UI) | N (UI) | |
| **a) Arm of randomization** | | | | | |
| Routine | 537 | - | 268 | 269 | RCT study logs |
| 2wT | 547 | - | 273 | 274 | RCT study logs |
| **b) Post-operative counselling—Day 0** | | | | | |
| Enrolled Nurse wage (per month) | - | $1,205 (602.50 to 1,807.50) | $1,205 (602.50 to 1,807.50) | $1,205 (602.50 to 1,807.50) | NDoH pay structure |
| Nurse counselling time in minutes (per patient)—Routine | - | 5 (2.5 to 7.5) | 5 (2.5 to 7.5) | 5 (2.5 to 7.5) | Assumption |
| Nurse counselling time in minutes (per patient) - 2wT | - | 10 (5 to 15) | 10 (5 to 15) | 10 (5 to 15) | Assumption |
| **c) Text service for 2wT patients** | | | | | |
| SMS aggregated set-up cost (per month) | - | $74.26 (37.13 to 111.39) | $74.26 (37.13 to 111.39) | $74.26 (37.13 to 111.39) | Africa is Talking |
| Cost per SMS | - | $0.02 (0.01 to 0.03) | $0.02 (0.01 to 0.03) | $0.02 (0.01 to 0.03) | Africa is Talking |
| Mean number of automated texts (per patient)—outbox | - | 20.1 (10.05 to 30.15) | 17.9 (8.95 to 26.85) | 22.3 (11.15 to 33.45) | Africa is Talking |
| Mean number of manual texts (per patient)—outbox | - | 5.65 (2.83 to 8.48) | 4 (2 to 6) | 7.3 (3.65 to 10.95) | Africa is Talking |
| Mean number of texts (per patient)—inbox | - | 14.5 (7.25 to 21.75) | 13 (6.5 to 19.5) | 16 (8 to 24) | Africa is Talking |
| Duration of 2wT program (days) | - | 13 (6.5 to 19.5) | 13 (6.5 to 19.5) | 13 (6.5 to 19.5) | RCT study logs |
| **d) Follow-up visits** | | | | | |
| Average number of visits attended—Routine | - | 1.34 (0.67 to 2.01) | 1.26 (0.63 to 1.89) | 1.42 (0.71 to 2.13) | RCT study logs |
| Average number of visits attended - 2wT | - | 0.22 (0.11 to 0.33) | 0.14 (0.07 to 0.21) | 0.3 (0.15 to 0.45) | RCT study logs |
| Unnecessary visits (Routine vs. 2wT) | | 1.12 (0.56 to 1.68) | 1.12 (0.56 to 1.68) | 1.12 (0.56 to 1.68) | RCT study logs |
| Administrator reception time in minutes (per patient) | - | 1.34 (0.67 to 2.00) | 0 | 2.67 (1.34 to 4.01) | Time-motion study |
| Nurse review time in minutes (per patient) | - | 5.43 (2.71 to 8.14) | 3.79 (1.90 to 5.69) | 7.06 (3.53 to 10.59) | Time-motion study |
| Administrator wage (per month) | - | $319 (159.5 to 478.5) | 0 | $638 (319 to 957) | NGO pay structure |
| Nurse wage (per month) | - | $1676.5 (838.25 to 2514.75) | $1,205 (602.50 to 1,807.50) | $2,148 (1,074 to 3,222) | NGO pay structure |
| Round trip distance to patient home (km) | - | - | 39.64 (19.82 to 59.46) | - | Time-motion study |
| Litres of petrol (per km) | - | 0.06 (0.03 to 0.09) | 0.06 (0.03 to 0.09) | - | Vehicle manual |
| Price of petrol (per litre) | - | $1.33 (0.67 to 2) | $1.33 (0.67 to 2) | - | Automobile Association |
| Transportation time cost in minutes (per visit) | - | 25.5 (12.75 to 38.25) | 34 (17 to 51) | - | Time-motion study |
| Transportation cost—nurse home visits (per visit) | - | $5.61 (2.81 to 8.42) | $7.48 (3.74 to 11.22) | $3.74 (1.87 to 5.61) | RCT study logs |
| Patients with AEs–Routine (%) | 0.01(0.005 to 0.014) | - | 0 | 0.02 (0.01 to 0.03) | RCT study logs |
| Patients with AEs - 2wT (%) | 0.02 (0.01 to 0.03) | - | 0.01(0.005 to 0.014) | 0.03 (0.02 to 0.05) | RCT study logs |
| Material cost per AE (e.g., antiseptic ointment) | - | $5.28 (2.64 to 7.92) | $5.28 (2.64 to 7.92) | $5.28 (2.64 to 7.92) | RCT study logs |
| **e) Tracing—Phone calls** | | | | | |
| Phone call service (per minute) | - | $0.08 (0.04 to 0.12) | $0.08 (0.04 to 0.12) | $0.08 (0.04 to 0.12) | Public records |
| Time cost per call (minutes) | - | 5 (2.5 to 7.5) | 5 (2.5 to 7.5) | 5 (2.5 to 7.5) | Assumption |
| Number of phone call attempts in tracing—Routine and 2wT | - | 3(1.5 to 4.5) | 3(1.5 to 4.5) | 3(1.5 to 4.5) | RCT study logs |
| Patients missed day 2 & day 7 visits—Routine (%) | 0.05 (0.03 to 0.08) | - | 0.02 (0.01 to 0.03) | 0.09 (0.05 to 0.14) | RCT study logs |
| Patients without post MC contact - 2wT (%) | 0.09 (0.05 to 0.14) | - | 0.06 (0.03 to 0.09) | 0.11 (0.06 to 0.17) | RCT study logs |

(*Continued*)

 

**Table 2.** (Continued)

| Parameters | Sum | Average | Rural | Urban | Source |
|---|---|---|---|---|---|
| | N (Uncertainty Interval) | N (UI) | N (UI) | N (UI) | |
| **f) Tracing—Home visits** | | | | | |
| Mean distance to patient home (km) | - | 29.73 (14.87 to 44.60) | 39.64(19.82 to 59.46) | 28.54 (14.27 to 42.81) | Time-motion study |
| Litres of petrol (per km) | - | 0.06(0.03 to 0.09) | 0.06(0.03 to 0.09) | 0.06 (0.03 to 0.09) | Vehicle manual |
| Price of petrol (per litre) | - | $1.33(0.67 to 2) | $1.33(0.67 to 2) | $1.33(0.67 to 2) | Automobile Association |
| Traced patients with no post MC contact—Routine (%) | 0.03 (0.02 to 0.05) | - | 0.03 (0.02 to 0.05) | 0.03 (0.02 to 0.05) | RCT study logs |
| Traced patients with no post MC contact - 2wT (%) | 0.002 (0.001 to 0.003) | - | 0.002 (0.001 to 0.003) | 0.002 (0.001 to 0.003) | RCT study logs |

data and costing algorithms. All parameters can be adjusted to reflect new inputs and scenarios, providing a tool for future costing studies of 2wT impact in other settings, contexts, and salary structures. All costs were collected in the South African rand (ZAR) and converted into the United States dollar (USD) using the exchange rate ZAR 1 = $0.066.

**Counselling.** Each VMMC client had a counselling session directly following circumcision. The post-operative counselling cost was estimated by the average time in minutes for a counselling session and wage per minute for the nurse performing the counselling.

Counselling cost = time * wage

**SMS follow-ups.** There were three components in SMS follow-ups per client. The first component was the 'Africa is Talking' SMS aggregator monthly set-up costs. The second component was the SMS service cost, i.e., the product of SMS unit cost and average number of SMS per client. The third component was the associated personnel cost of sending a manual SMS.

SMS cost = set-up cost + SMS service cost + manual SMS time cost

## Follow-up visits and AE management

The rural costing model included the estimation of transportation fuel cost, transportation time cost, and nurse review time cost per visit. In urban settings, we estimated reception time cost and nurse review time cost per visit. We used the average number of follow-up visits per client by location. Transportation cost was paid by the program in rural settings whereas patients paid for transportation in urban settings. In the rural setting, an enrolled nurse interacted with clients via 2wT and usually drove out to meet clients at their homes or workplaces to conduct any requested post-VMMC reviews. In urban settings, clients interacted with clients via 2wT and typically returned to the clinic on their own for visits when needed or desired. Reception time cost only applied to urban settings as the rural settings did not have reception.

Cumulative moderate and severe AEs as defined by global VMMC standards [33] were reported from day 1 through day 21 by randomization group and by location. The time costs in treating AEs were included in overall follow-up visits. We estimated the material costs for AE management separately.

Rural visit cost = # of visits * (transportation fuel cost + transportation time cost + review time cost)

Urban visit cost = # of visits * (reception time cost + review time cost)

In rural settings, transportation fuel cost = Round trip distance to client home (km) * Litres of petrol (per km) * Price of petrol (per litre)

AE management = probability of AE * AE material cost

**Tracing.**   In routine care, if the client missed both day 2 and day 7 visits, three phone calls were attempted to confirm healing. In the 2wT arm, if the client had no SMS contact by day 8, the client was actively traced via phone up to three attempts. Successful calls to clients in both arms took about 5 minutes If the client was not reachable by phone, up to three home visits were attempted for both study arms. The number of clients eligible for phone tracing and home tracing was reported in the RCT [23].

Phone tracing cost = probability of phone tracing * (phone call time cost in up to 3attempts + phone call service cost of a successful completed call).

For home tracing, transportation cost was estimated in the care model that the nurse drove to the client's home to conduct the review. Transportation fuel cost, transportation time cost, and nurse review time cost were also included. We considered the least expensive model, in which the nurse also served as the driver. The urban tracing distance is 72% of the distance in rural settings according to the survey results in our RCT [23].

Home tracing cost = probability of home tracing * (transportation time cost in all attempts + transportation fuel cost in all attempts + review time cost in successful attempt).

For all tracing attempts, phone service costs and client review costs were only recorded for the successful attempt. For instance, if the nurse made three home visits and the client was reached in the last visit, the client review cost for only the one successful visit was included.

## Ethics

This Multiple Principal Investigators (MPI) study was approved by the Internal Review Boards of the University of Washington (Study 00009703, PI: Feldacker) and the University of the Witwatersrand, Human Research Ethics Committee (Ethics Reference No: 200204, PI: Setswe). All RCT participants provided written informed consent for use of individual-level data collected from both 2wT study specific sources and routine VMMC program data collection. All data obtained and utilized for the costing study was de-identified data from study databases and contained no identifying information. As part of the overall RCT IRB approvals, a waiver of consent was granted for healthcare worker observations from the time-motion study.

## Results

### VMMC clients

A total of 1,460 VMMC clients were recruited for the RCT, and 141 (9.7%) clients were found ineligible for 2wT: 103 (45.2%) had no cell phones, 21 (5.7%) had a language barrier, 13 (3.5%) could not read or write, 2 (0.5%) were blind, and 2 (0.5%) were unfit to consent) [23]. The costing analysis included all 1,084 enrolled clients from the RCT: 537 clients in routine care and 547 clients in 2wT intervention. Table 2 indicates key costing parameters stratified by urbanicity. Average costs per client by randomization group, cost savings, and savings by location are presented in Table 3, with illustrations in Figs 1 and 2. Costing scenarios are presented in Table 4.

### Counselling

In both settings, an enrolled nurse with a monthly salary of $1,205 was assumed to spend 5 minutes in routine post-operative counselling and 10 minutes in 2wT counselling (Table 2). There was no difference in counselling time by location.

**Table 3. Unit cost for routine care (standard of care (SoC) and 2wT ($USD).**

| Costs per patient | Rural | | | Urban | | | Average Overall | | |
|---|---|---|---|---|---|---|---|---|---|
| | SoC | 2wT | 2wT vs. SoC | Routine | 2wT | 2wT vs. SoC | SoC | 2wT | SoC vs. 2wT |
| Post-op counselling—day 0 | $0.63 | $1.26 | $0.63 | $0.63 | $1.26 | $0.63 | $0.63 | $1.26 | $0.63 |
| Two-way texting | $0.00 | $1.05 | $1.05 | $0.00 | $2.30 | $2.30 | $0.00 | $1.67 | $1.67 |
| Set-up cost | $0.00 | $0.06 | $0.06 | $0.00 | $0.06 | $0.06 | $0.00 | $0.06 | $0.06 |
| SMS service cost | $0.00 | $0.49 | $0.49 | $0.00 | $0.61 | $0.61 | $0.00 | $0.55 | $0.55 |
| Manual SMS time cost | $0.00 | $0.50 | $0.50 | $0.00 | $1.63 | $1.63 | $0.00 | $1.07 | $1.07 |
| Follow-up visits | $10.04 | $1.12 | -$8.92 | $2.50 | $0.53 | -$1.97 | $6.26 | $0.82 | -$5.44 |
| Provider transportation fuel cost | $4.05 | $0.45 | -$3.60 | $0.00 | $0.00 | $0.00 | $2.02 | $0.23 | -$1.80 |
| Provider transportation time cost | $5.39 | $0.60 | -$4.79 | $0.00 | $0.00 | $0.00 | $2.69 | $0.30 | -$2.39 |
| Provider review time cost | $0.60 | $0.07 | -$0.53 | $2.50 | $0.53 | -$1.97 | $1.55 | $0.30 | -$1.25 |
| AE treatment (materials) | $0.00 | $0.04 | $0.04 | $0.10 | $0.17 | $0.08 | $0.05 | $0.11 | $0.06 |
| Tracing—Phone calls | $0.03 | $0.19 | $0.16 | $0.30 | $0.39 | $0.09 | $0.16 | $0.29 | $0.12 |
| Tracing—Home visits | $0.73 | $0.04 | -$0.69 | $0.56 | $0.03 | -$0.53 | $0.64 | $0.04 | -$0.61 |
| **Total costs** | $11.42 | $3.69 | -$7.73 | $4.08 | $4.67 | $0.59 | $7.75 | $4.18 | -$3.56 |

## SMS follow-ups

The SMS aggregator set-up cost charged by 'Africa is Talking' was $74.25 per month for the 2wT system as detailed by Table 2. Each client was sent automated text messages for 13 days in a month of 30 days and the average cost per client was proportioned to 43% of the month. With 547 clients in 2wT, the set-up cost per client was $0.14. When 'Africa is Talking' serves more clients in future, the set-up cost is expected to reduce.

Table 2 also indicates that a standard SMS of 160 characters cost $0.02. Each 2wT client received 20.1 (10.05 to 30.15) messages on average, including 5.65 (2.83 to 8.48) manual messages from nurses. Clients sent an average of 14.5 (7.25 to 21.75) messages, total, between day 1 and Day 13 to report daily status and any AE concerns. Urban clients were more active in 2wT, sending an average of 3 more messages than rural clients. Similarly, there were 3 more messages from the nurse on average, per client, in urban settings than in rural settings.

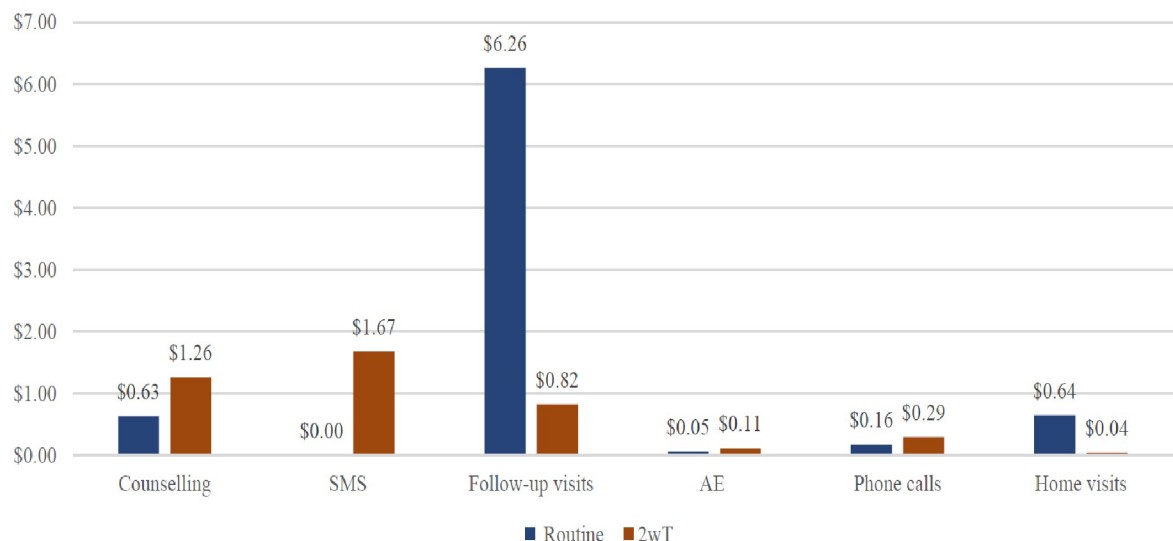

**Fig 1. Costs in USD for routine care and 2wT per client (USD).**

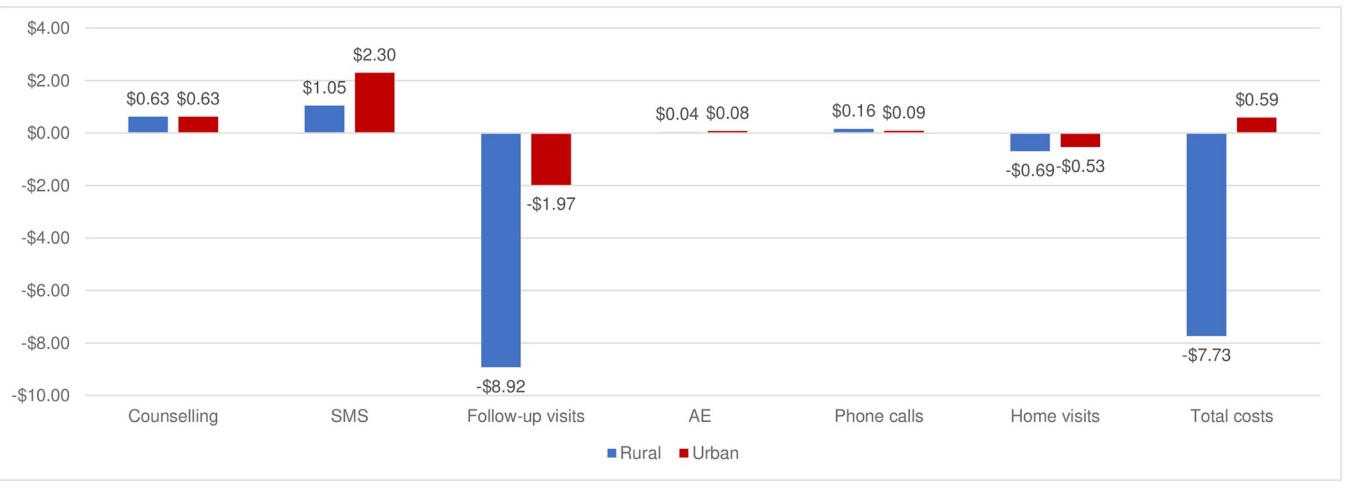

**Fig 2. Cost changes by implementing 2WT compared to the routine, rural vs. urban (USD).**

### Follow-up visits and adverse event management

**Clinic visits.** As previously reported in the RCT, and as expected per RCT intervention assignment, routine clients had more clinic follow-up visits than 2wT clients. Routine clients had 1.34 (0.67 to 2.01) visits on average between day 2 and day 13: 1.26 (0.63 to 1.89) in rural areas and 1.42 (0.71 to 2.13) in urban locations. 2wT clients had 0.22 (0.11 to 0.33) visits on average, with 0.14 (0.07 to 0.21) in rural areas and 0.3 (0.15 to 0.45) in urban clinics. Urban clients, in both routine and 2wT groups, had more visits than rural clients. Post-operative engagement in care was higher among 2wT arm than control: 94% of 2wT males responded to at least one 2wT message and 80.4% of control arm males attended at least 1 in-person, post-operative visit [23].

**AE material cost.** The study identified a total of 16 AEs across both arms. Among 719 visits in the routine arm, there were 5 AEs identified whereas 11 AEs were identified among 118 visits in 2wT arm. Material cost per AE, including bandage, paraffin gauze, and antiseptic ointment, was estimated at $5.28.

**Salary and time cost.** In the rural setting, Enrolled Nurses with a monthly salary of $1,205 conducted the post-VMMC reviews with 3.79 (1.90 to 5.69) minutes per client. In urban settings, administrators with a monthly salary of $638 facilitated reception with 2.67 (1.34 to 4.01) minutes per client and a *Professional Nurse* with a monthly salary of $2,148 conducted the reviews with 7.06 (3.53 to 10.59) minutes per client. The routine urban *Professional Nurse* salary was 1.78 higher than the rural Enrolled Nurse salary.

**Transportation cost.** In rural settings, routine VMMC teams drove to review clients or provided transport while urban clients were largely responsible for their own transportation. It was estimated that the round–trip distance to a client's home on average was 39.64 (19.82–9.91) kilometers in rural areas. As indicated in the vehicle manual, the average fuel

**Table 4. Potential cost savings from 2wT (USD).**

| Cost saving (2wT vs. Routine) | Rural | Urban | Average |
|---|---|---|---|
| **Routine—One visit** | -$5.66 | $1.33 | -$2.16 |
| **Routine—1.34 visits (observed)** | -$7.73 | $1.33 | -$3.56 |
| **Routine—Two visits** | -$13.63 | -$0.43 | -$7.02 |

consumption of the vehicle that was used for tracing was 0.06L/km (*6L/100km*). Information from the Automobile Association reported the fuel price of $1.33/L [34]. The transportation fuel cost was $3.21 (1.61 to 4.82) in rural areas.

From time-motion data, it was estimated that the program's transportation time per client visit was 34 minutes in rural settings. No client transportation was provided by the program in urban settings over the observation period.

### Tracing

Among routine men, 29 clients did not attend any visit and were potentially lost-to-follow-up (LTFU). Potential LTFU were traced first by phone; 17/29 were not reached by phone and were traced by home visit. For the participants in the 2wT arm, 47 had no contact by day 8 post-VMMC; 46 were reached by phone and 1 was traced by home visit.

The cost of phone call was $0.08 per minute. It was assumed that an Enrolled Nurse conducted the phone calls, and an average phone call time was 5 minutes. To trace potential LTFU clients, it was assumed that on average 3 phone calls were attempted in both routine group and 2wT group. In home tracing, the transportation fuel cost was the same as the clinic visit. The estimated tracing transportation costs on average, including fuel and personnel costs, were $0.64 and $0.04 in rural and urban areas, respectively. According to our time-motion study, it is important to note that home visits were costly both in terms of money and time, especially in rural areas. The average travel time to reach one client was 34 minutes (range: 17 to 51 minutes) and the average review time was 3.8 minutes (range: 1.9 to 5.7 minutes). On a typical day of rural tracing, the nurse spent 90% of the time driving. On average, 5 clients (range: 2.5 to 7.5 clients) would be traced in one day in rural settings.

### Costs and cost savings

Table 2 shows that there were, on average, 1.12 unnecessary post-operative visits per person, reflecting the difference between average number of in-person visits in the control arm (1.34) and average number of elected visits in the 2wT arm (0.22). In the 2wT arm, visit attendance reflects demand for in-person, post-VMMC care, 0.14 and 0.3 visits in rural and urban areas, respectively.

Table 3 shows that the average cost per post-operative visit was $7.75 in routine care and $4.18 in 2wT. The cost saving of $3.56 was primarily seen in rural areas ($7.73), with no savings in urban areas (with $0.59 increased cost). Table 3 also shows the disaggregation of costs by location. The average post-operative counselling cost was $0.63 and $1.26 in routine and 2wT groups, respectively. SMS costs were only incurred among the 2wT group, with $1.67 average costs, of which $2.30 was in urban areas and $1.05 in rural areas.

Fig 1 illustrates the cost distribution across the post-VMMC care continuum. In the 2wT group, the costs were concentrated in early actions within the care continuum, including counselling ($1.26) and SMS ($1.67), which accounted for 70% of the overall costs. By contrast, in the routine group, most of the financial investment was in clinical visits, $6.26 out of $7.75 (81%), followed by counselling ($0.63). In the post-VMMC care continuum, 2wT intervention shifts the healthcare effort to preparing the client to actively engage in their healing process and to encourage them to promptly seek care via consistent daily communication with the client from day 1 to day 13.

Fig 2 demonstrates costs and savings by urbanicity. Overall, there were $0.59 cost increases in urban areas and $7.73 savings in rural area by implementing 2wT instead of utilizing routine care. Cost savings were from reduced follow-up visits ($1.97 in urban areas and $8.92 in rural areas). Savings from home visits in tracing were $0.53 in urban areas and $0.69 in rural areas. Increased costs in 2wT arm by location included counselling ($0.63, the same in each

location), SMS ($2.30 in urban areas and $1.05 in rural areas), AE materials ($0.08 in urban areas and $0.04 in rural areas), and calls ($0.09 in urban areas and $0.16 in rural areas).

## Other scenarios

Table 4 depicts three scenarios, by location, in comparing cost savings of using 2wT compared to routine care visits namely one visit, 1.34 visits on average, and two visits within routine care. If there had been one required visit per client in routine care, the average cost saving was calculated at $2.16 per client. In the scenario drawn from the pragmatic RCT of 1.34 visits within routine care, the average cost saving was $3.56. If two follow-up visits were required per client in routine, the average cost saving was $7.02 per client. An addition scenario with Enrolled Nurse salary applied to both rural and urban settings had little impact, with average overall savings of $3.14 using 2wT over routine with the observed visit patterns, showing the robustness of these savings over modest salary changes.

## Discussion

In this costing analysis, we determined that 2wT-based VMMC follow-up saved an average of $3.56 per client across settings, with savings ranging from $2.16 to $7.02. 2wT improves the quality of post-VMMC follow-up care at lower overall cost by providing an SMS-based mHealth option for clients with access to cell phones, encouraging visits when needed instead of compulsory visits on day 2 and/or day 7. Savings using the 2wT approach were higher in rural as compared to urban areas–an important finding as the majority of VMMC program implementation occurs in rural areas [14]. Even though using 2wT costs slightly more in urban areas, the better quality of post-VMMC follow-up care justifies these additional costs. Among 2wT costs, the largest costs were for the SMS, themselves, an investment to actively engage men into quality follow-up care with a 2wT nurse for reassurance or triaging to in-person review when needed. The 2wT clients on average received 20.1 text messages and sent 14.45 text messages in the post-operative period for an SMS cost of $1.67 per client. In urban areas, males interacted more via 2wT, raising costs but also increasing direct client-to-clinician communication. Cost savings using 2wT as compared to routine post-operative care were similar in the 2wT RCT conducted in Zimbabwe that found a net savings of $2.10 from the payer (Ministry of Health or donor) perspective [14]. As attendance at routine post-operative visits may be overreported in VMMC programs operating at scale, cost savings and benefits for quality follow-up care using the 2wT approach may be underestimated.

Cost savings from 2wT vary by adherence to required attendance at post-VMMC visits and by location. Given that moderate and severe AE rates in male circumcisions are low (1%-2%) [4], it is costly to require all VMMC clients to attend one or two post-operative in-person clinic visits. 2wT can save an average of $7.02 ($13.63 savings in rural and $0.43 savings in urban) compared to the routine care with two required visits. Our study shows that routine follow-up in rural areas was expensive as the district geography is expansive. For scheduled day 2 and day 7 post-operative visits, the CHAPS rural VMMC team of a nurse and a dedicated driver, arranged to meet the clients at a convenient community location for physical client reviews or a client contacted the CHAPS VMMC team for pickup or an in-person home visit, with occasional clients returning independently to the VMMC site. In contrast, the urban model had three teams based at three static clinics. Urban clients were expected to commute for review visits, independently, at the dedicated static sites.

2wT may increase some costs that are likely offset by additional quality care benefits. Firstly, enhanced counselling incurred a minimal cost of $0.63, but enabled the clients to effectively identify and communicate AE concerns in daily SMS–a worthwhile cost. Second, daily SMS

communication improved early detection of AEs and subsequent swift referral of those in need for in-person clinical visits. This triaging process led to identification of AEs earlier with less severity, likely averting costs of more severe AEs [23]. Third, only the clients with AE concerns or those desiring reassurance attended clinic visits, resulting in only 0.22 elected visits per 2wT client (Table 3), a reduction in review workload. Lastly, reduced follow-up visits could allow providers to concentrate on cases in need of medical attention and free up time to commence surgery early and on-time, potentially increasing client satisfaction.

Quality assurance likely also benefits from 2wT improvements in verification of, and supervision for, quality post-operative care. For NDoH, 2wT follow-up adhered to government regulations regarding client privacy and complied with required NDoH documentation for VMMC client follow-up. The 2wT system provides verification of timely, nurse-led, post-operative support in line with PEPFAR guidelines, facilitating quality follow-up regardless of client location. 2wT documentation also provides another layer of quality assurance to confirm program performance. Unique client verification via the 2wT system increases confidence in program productivity, providing another data source to prevent duplication in reporting. Although not formally considered in the costing analysis, supervision costs of 2wT may be lower than those for routine care. Routine monitoring of care quality via review of paper forms is time-consuming, requiring intensive efforts. However, 2wT dashboards and system-embedded hierarchies allow for remote client oversight and access to data for timely program monitoring. Supervisors can access 2wT messages, reports, tasks, and client records from multiple sites or remotely from a central location. Similarly, quality assurance activities can also take place virtually, allowing managers and program administrators to review data and provide clinical oversight, informing improvements. In future, calculation of these costs may demonstrate further benefit of the 2wT approach for routine VMMC follow-up.

## Limitations

There are several assumptions and limitations in this study. First, this study focuses on the cost of the activities to serve an average client presuming that 2wT is implemented within an existing routine VMMC service. Therefore, start-up costs (e.g., developing text message library, translation to local languages, adaptation to South Africa context) and fixed costs (e.g., full-time employment dedicated to 2wT, cell phones, computers, vehicle purchase, insurance, maintenance cost) were excluded. Second, we conducted costing from the perspective of the VMMC program and current guidelines to inform feasibility of adoption by the NDoH, assuming the existing VMMC donors would continue support for at least the 5-year time frame. However, with any payer (Ministry of Health, donor, or program), and evolving guidelines, 2wT costs and comparators could change over time. Third, we used the standard workweek in South Africa, i.e., 5 days per week, from Monday to Friday. In calculating hourly wage, we used the salary of the Enrolled Nurse cadre and did not take into account that some nurses work on weekends and public holidays or may have additional days off, such as annual leave or sick leave–considerations that could affect costs. For AE management, only costs related to reportable moderate and severe cases are documented and noted in this study. We did not measure the costs to manage mild AEs such as pain. Lastly, we explored will see savings (monetary cost) in reduced transportation fuel costs that the reduced staff time required for follow-up (an opportunity cost not a monetary cost savings) means that overall monetary costs to the program are likely to increase.

## Conclusion

Evidence from two RCTs in Zimbabwe and South Africa demonstrate that this 2wT approach provides high-quality VMMC follow-up as compared to required in-person reviews and

lowers overall VMMC program costs. Rural savings using 2wT offset nominal increased costs in urban areas. Additional 2wT associated improvements in care quality, supervision, and verification also likely leading to longer-term savings. The health sector should invest in 2wT. In the context of National VMMC targets of 350,000 to 500,000 VMMCs per annum, employing 2wT could dramatically reduce the number of in-person post-operative reviews, resulting in concrete efficiency gains and significant annual program savings. Investing these resources back into the VMMC program could further expand VMMC access, improve care quality, and advance VMMC program goals of safe, efficient, and effective VMMC scale-up.

## Supporting information

**S1 File. 2wT costing study EpiCollect form.** Tool used for 2WT time-in-motion client follow-up data collection.
(PDF)

**S1 Table. 2wT costing tool.** Modifiable Excel spreadsheet for 2wT scenario costing.
(XLSX)

## Acknowledgments

The authors would like to thank the following: the Departments of Health of the Gauteng and Northwest Provinces, Bojanala district and the Ekurhuleni Health District Research Committee (EHDRC) for allowing us to conduct the study in their districts; implementing partner, Right to Care and the CHAPS study implementation team. The authors would like to thank the Medic team and all study participants for their involvement in the study. The authors would like to thank Lingchao Mao and Ziwei He for independent replications of the results as well as thank Simon Ding and Emily Chu for their comments.

## Author Contributions

**Conceptualization:** Yanfang Su, Jacqueline Pienaar, Geoffrey Setswe, Caryl Feldacker.

**Formal analysis:** Yanfang Su, Rachel Mukora, Xinpeng Xu, Hannock Tweya.

**Funding acquisition:** Jacqueline Pienaar, Geoffrey Setswe, Caryl Feldacker.

**Investigation:** Jacqueline Pienaar, Geoffrey Setswe, Caryl Feldacker.

**Methodology:** Yanfang Su, Rachel Mukora, Felex Ndebele, Jacqueline Pienaar, Calsile Khumalo, Hannock Tweya, Maria Sardini, Geoffrey Setswe, Caryl Feldacker.

**Resources:** Geoffrey Setswe, Caryl Feldacker.

**Supervision:** Yanfang Su, Geoffrey Setswe, Caryl Feldacker.

**Writing – original draft:** Yanfang Su, Rachel Mukora, Felex Ndebele, Calsile Khumalo, Geoffrey Setswe, Caryl Feldacker.

**Writing – review & editing:** Yanfang Su, Rachel Mukora, Felex Ndebele, Jacqueline Pienaar, Calsile Khumalo, Hannock Tweya, Maria Sardini, Sarah Day, Kenneth Sherr, Geoffrey Setswe, Caryl Feldacker.

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
