## [Decision Letter · Decision Letter 0]

3 Mar 2023

PONE-D-23-01844Cost savings in male circumcision post-operative care continuum in rural and urban South Africa: Evidence on the importance of initial counselling and daily SMSPLOS ONE

Dear Dr. Su,

Thank you for submitting your manuscript to PLOS ONE. After careful consideration, we feel that it has merit but does not fully meet PLOS ONE’s publication criteria as it currently stands. Therefore, we invite you to submit a revised version of the manuscript that addresses the points raised during the review process. As you will note, the comments provided by two reviewers are significantly more substantial than the third. I ask that you carefully review all comments and either edit the paper accordingly or provide a thoughtful, rationale rebuttal to any comments with which you disagree.

We look forward to receiving your revised manuscript.

Kind regards,

Lisa Suzanne Dulli, PhD

Academic Editor

PLOS ONE

Journal Requirements:

"Research reported in this publication was supported by the National Institute of Nursing Research (NINR) of the National Institutes of Health under award number 5R01NR019229, “Expanding and Scaling Two-way Texting to Reduce Unnecessary Follow-Up and Improve Adverse Event Identification Among Voluntary Medical Male Circumcision Clients in the Republic of South Africa.”

"Research reported in this publication was supported by the National Institute of Nursing Research (NINR) of the National Institutes of Health under award number 5R01NR019229, “Expanding and Scaling Two-way Texting to Reduce Unnecessary Follow-Up and Improve Adverse Event Identification Among Voluntary Medical Male Circumcision Clients in the Republic of South Africa.” The content is solely the responsibility of the authors and does not necessarily represent the official views of  the National Institutes of Health."

Reviewers' comments:

Reviewer's Responses to Questions

**Comments to the Author**

1. Is the manuscript technically sound, and do the data support the conclusions?

Reviewer #1: Partly

Reviewer #2: Yes

Reviewer #3: No

2. Has the statistical analysis been performed appropriately and rigorously? 

Reviewer #1: N/A

Reviewer #2: I Don't Know

Reviewer #3: No

3. Have the authors made all data underlying the findings in their manuscript fully available?

Reviewer #1: No

Reviewer #2: Yes

Reviewer #3: No

4. Is the manuscript presented in an intelligible fashion and written in standard English?

Reviewer #1: Yes

Reviewer #2: Yes

Reviewer #3: No

5. Review Comments to the Author

Reviewer #1: General Comments

The authors are to be commended for designing and completing an rigorous study comparing alternative ways to handle VMMC post-op care. While the main finding of reduced use of in-person health services associated with the two-way texting (2wT) compared to routine in-person follow-up is not in question, there is some concern about who realizes the cost-savings and the implications for the health sector. As shown in Table 4, the bulk of the cost savings from 2wT comes from reduced follow-up visits (0.22 per person in the 2wT cohort (line 254) vs. 1.34 in the routine cohort (line 252)). Yet, these differences are more likely to result in financial (transport) and time savings (opportunity costs) to the clients than cost savings to the health system. I understand your attempt to monetize these as savings to the health system by considering three hypothetical scenarios, but the fact remains that these will be savings to the client and therefore it is disingenuous to talk about these as savings to the health system as you do in the conclusion. In fact, the savings within the health system are most likely going to be labor time savings (opportunity costs) and not monetary savings due to reduced use of consumable supplies. If you remove the cost of transport for follow-up visits (~$3.91 from line 287), then there would no longer be a monetary savings to the health sector. This does not mean that this type of intervention is not worth supporting as it does not lead to worse outcomes and is certainly more convenient to the clients while at the same time reducing the workload burden within the health system.

I would encourage the authors to consider presenting the results with stratification as to who bears the cost (health system or clients) rather than create the hypothetical scenario where client costs are loaded onto the health system. This would make the analysis more transparent and avoid the trap of thinking of the savings as monetary savings when the majority are opportunity cost savings as current health care staff are now available to address other needs.

Specific Comments:

Line 327: Where does the $7.6 (76%) figure come from. Looking at Table 4, should the denominator be $6.48 and the percentage 83%? Also, it is customary to report USD with 2 decimal places (applies throughout text and figure 1.

Table 4, can you include “$” in the Total cost row so it is clear that these are all USD amounts?

Fig 1: Please add a vertical axis title making it explicit that these are USD amounts in the bars.

Fig 2: Please add a vertical axis title making it explicit that these are USD amounts in the bars.

Table 5: Label second data row as Routine – 1.34 visits (observed).

Lines 387 – 394: This is where I think you are overselling your results. These will not be monetary savings that can be reprogramed to different uses. Most will be time savings for existing staff which can be repurposed as discussed in the preceding paragraph.

Lines 414-416: Rephrase this once you know where the cost savings occur (health sector or clients).

Reviewer #2: This is a well-conducted costing assessment of an SMS-based alternative to routine follow-up care for VMMC compared with standard of care, which involves several return clinic visits. The intervention is relevant and well described, and the need to assess cost and potential cost-savings is also well established. I am not an expert in costing analyses and certainly suggest that at least one reviewer be selected who is such an expert. However, based on my reading it seemed to be both well conducted and clearly presented. I found the overall manuscript to be well written and appropriately justified in each section. I did see one or two potential wording edits – for example, I would suggest that all results be reported in the past tense. However, I overall believe this is a strong manuscript and support its publication.

Reviewer #3: Please see my reviewer report attached separately. It will be easier to reach that document as formatted rather than writing more here.

Please see my reviewer report attached separately. It will be easier to reach that document as formatted rather than writing more here.

6. PLOS authors have the option to publish the peer review history of their article (what does this mean?). If published, this will include your full peer review and any attached files.

Reviewer #1: No

Reviewer #2: **Yes: **Caitlin Kennedy

Reviewer #3: No

---

## [Author Response · Author response to Decision Letter 0]

30 Jun 2023

We thank the editor and reviewers for reviewing this paper and providing constructive comments and suggestions. Point-to-point response to each comment is provided in the attached document, and corresponding changes in the manuscript are tracked. 

Best,

Yanfang, Rachel, Caryl and co-authors

---

## [Decision Letter · Decision Letter 1]

21 Aug 2023

PONE-D-23-01844R1Cost savings in male circumcision post-operative care using two-way texting mHealth in rural and urban South AfricaPLOS ONE

Dear Dr. Su,

Thank you for submitting your manuscript to PLOS ONE. After careful consideration, we feel that it has merit but does not fully meet PLOS ONE’s publication criteria as it currently stands. Therefore, we invite you to submit a revised version of the manuscript that addresses the points raised during the review process.

Only one of the previous reviewers was available to review your revisions. With my own review, it seems as though many of the comments submitted by the other reviewers were either largely addressed or you were able to explain your decision not to address the points. However, I do agree with the first reviewer that the results of the study are not being adequately contextualized and are even being "oversold" as to their importance. The manuscript does hold merit if the authors are able to make some few remaining adjustment. Specifically, the discussion section begins with a statement regarding the average cost-savings per client in this study. Two very important points are not well addressed. First, nearly all the costs savings were seen in the rural intervention setting. In fact, the 2wT intervention was more costly than routine follow-up in the urban setting. Yet, your conclusion begins "With evidence from two RCTs in Zimbabwe and South Africa demonstrating 2wT safety and cost benefits in urban and rural contexts, the health sector should invest in 2wT." Results from this study do not demonstrate a cost benefit in urban settings, and recommendations to invest in the intervention would only benefit programs with a larger rural client population, based on costs alone.  Your estimates of the potential cost benefits are predicated on your findings that included a sample with equal numbers of participants in both the urban and rural settings for both study arms. This condition needs to be made clear. Second, as one reviewer pointed out, the majority of savings are from follow-up visits and driven predominately by cost-savings in provider time (Table 3). As the first reviewer noted, "These will not be monetary savings that can be reprogramed to different uses. Most will be time savings for existing staff which can be repurposed..."  This also needs to come out more clearly in the text. Both of these points will likely be key considerations for a country or a program that might consider this intervention. Lastly, one smaller point noted by review 1 is that both figures are missing axis titles. Please add so the figures are able to stand alone and be completely understood.

We look forward to receiving your revised manuscript.

Kind regards,

Lisa Suzanne Dulli, PhD

Academic Editor

PLOS ONE

Reviewers' comments:

Reviewer's Responses to Questions

**Comments to the Author**

1. If the authors have adequately addressed your comments raised in a previous round of review and you feel that this manuscript is now acceptable for publication, you may indicate that here to bypass the “Comments to the Author” section, enter your conflict of interest statement in the “Confidential to Editor” section, and submit your "Accept" recommendation.

Reviewer #1: (No Response)

2. Is the manuscript technically sound, and do the data support the conclusions?

Reviewer #1: Partly

3. Has the statistical analysis been performed appropriately and rigorously? 

Reviewer #1: Yes

4. Have the authors made all data underlying the findings in their manuscript fully available?

Reviewer #1: Yes

5. Is the manuscript presented in an intelligible fashion and written in standard English?

Reviewer #1: Yes

6. Review Comments to the Author

Reviewer #1: General Comments:

Thank you for the clarification that the PEPFAR supported programs are covering the transport costs for clients in the rural area. Also, the emphasis on looking at costs from the payer’s perspective. However, I am still concerned that while the PEPFAR-funded program (CHAPS) will see savings (monetary cost) in reduced transportation fuel costs that the reduced staff time required for follow-up (an opportunity cost not a monetary cost savings) means that overall monetary costs to the program are likely to increase. A further concern is why provider time is being assigned to transportation. If providers are leaving the clinic to visit the client at their home or somewhere else in the community, that provider is accumulating a lot of time when they are not available for other clients. In a country with shortages of skilled personnel this seems like a poor use of human resources. Why not send a driver to pick the client and return them home while the auxiliary nurse remains at the clinic to assist other clients? As mentioned on line 321, on a day of rural tracing the nurse is spending 90% of the time driving.

I am still troubled by the conclusion that the use of 2wT could yield savings of over $1.1 million per year (line 442). As stated above, the labor savings are not monetary savings so there is no budgetary reduction associated with these “savings”. Also, I’m not sure there is evidence to suggest men’s hesitancy to take up VMMC because of the need to return for follow-up visits and with more efficient follow-up that uptake would increase (lines 444-446).

Specific Comments:

Line 214 & 215 (& perhaps elsewhere) replace “Aes” with “AEs”. My guess is your word processing software was trying to be helpful .

Line 242-244: The decision to only count the cost for a successful attempt even if multiple attempts were required seems to put downward bias on tracing costs. Please justify this decision.

Line 360: According to Line 113, NDoH protocols already require two post-surgery follow-up visits. So why is this being treated as a hypothetical scenario?

7. PLOS authors have the option to publish the peer review history of their article (what does this mean?). If published, this will include your full peer review and any attached files.

Reviewer #1: No

---

## [Author Response · Author response to Decision Letter 1]

1 Oct 2023

Response to reviewers

PONE-D-23-01844R1

Cost savings in male circumcision post-operative care using two-way texting mHealth in rural and urban South Africa

September 29, 2023

Dear Plos ONE Editorial Team,

Thank you very much for the opportunity to revise the manuscript again to comply with these additional helpful comments. We have reviewed and updated the text, responded to the remaining suggestions, and provided an explanation for outstanding issues which we hope will alleviate any remaining issues before consideration of publication. We respond in italics to the remaining issues and provided a track changes version to review additional revisions, edits, and updates for clarity throughout. 

Editor: I do agree with the first reviewer that the results of the study are not being adequately contextualized and are even being "oversold" as to their importance. The manuscript does hold merit if the authors are able to make some few remaining adjustment. Specifically, the discussion section begins with a statement regarding the average cost-savings per client in this study. Two very important points are not well addressed.

First, nearly all the costs savings were seen in the rural intervention setting. In fact, the 2wT intervention was more costly than routine follow-up in the urban setting. Yet, your conclusion begins "With evidence from two RCTs in Zimbabwe and South Africa demonstrating 2wT safety and cost benefits in urban and rural contexts, the health sector should invest in 2wT." Results from this study do not demonstrate a cost benefit in urban settings, and recommendations to invest in the intervention would only benefit programs with a larger rural client population, based on costs alone. Your estimates of the potential cost benefits are predicated on your findings that included a sample with equal numbers of participants in both the urban and rural settings for both study arms. This condition needs to be made clear. Second, as one reviewer pointed out, the majority of savings are from follow-up visits and driven predominately by cost-savings in provider time (Table 3). As the first reviewer noted, "These will not be monetary savings that can be reprogramed to different uses. Most will be time savings for existing staff which can be repurposed..." This also needs to come out more clearly in the text. Both of these points will likely be key considerations for a country or a program that might consider this intervention.

Response: 

Thank you for this opportunity to provide clarity on why the rural savings would lead to overall savings in the program. Under the Aurum and CHAPS program, VMMC services are delivered predominantly in rural areas, up to 90% of the patients are from rural settings in any month as published in our clinical JMIR paper. We now added this statement in several areas of the paper, including in paragraph 5 of the introduction, “The VMMC program in SA is predominantly implemented in more rural areas [14].” We also enhanced our discussion of the savings in rural areas and explained why we think that the minimal additional costs in urban areas are a worthy investment. 

We also softened and updated the language of the discussion about the cost savings, “2wT has additional benefits for cost savings that could lead to potential long-term cost-savings in post-VMMC care delivery. Firstly, enhanced counselling incurred a minimal cost of $0.63, but enabled the clients to effectively identify and communicate AE concerns in daily SMS – a worthy investment worthwhile cost. Second, daily SMS communication improved early detection of AEs and subsequent swift referral of those in need for in-person clinical visits. This triaging process led to identification of AEs earlier with less severity, likely averting costs of more severe AEs[14]. Third, only the clients with AE concerns or those desiring reassurance attended clinic visits, resulting in only 0.22 elected visits per 2wT client (Table 3), a reduction in review workload. Lastly, reduced follow-up visits allow providers to concentrate on cases in need of medical attention and free up time to commence surgery early and on-time, potentially increasing client satisfaction.”

Lastly, one smaller point noted by review 1 is that both figures are missing axis titles. Please add so the figures are able to stand alone and be completely understood.

Response: 

Completed. 

Reviewer #1: General Comments:

Thank you for the clarification that the PEPFAR supported programs are covering the transport costs for clients in the rural area. Also, the emphasis on looking at costs from the payer’s perspective. However, I am still concerned that while the PEPFAR-funded program (CHAPS) will see savings (monetary cost) in reduced transportation fuel costs that the reduced staff time required for follow-up (an opportunity cost not a monetary cost savings) means that overall monetary costs to the program are likely to increase. 

Response: 

These will be monetary savings that can be reprogramed to different uses, as we noted in the paper, like time savings for existing staff who would not be completed unnecessary reviews. In contrast, opportunity cost refers to the value of the next best alternative that one foregoes when making a decision, choosing one option always comes at the cost of not choosing another. In our study, we estimated time costs for health workers and monetize time costs by using the salary information. We didn’t estimate time savings for patients – a separate study that we did not conduct in large part as both services (VMMC) and transportation are largely provided by the program. Patient costs have been covered previously in the literature. Moreover, we look at the payer perspective since, as the reviewer notes, the provider time is being assigned to transportation – which is a wasted resource. While we agree, and we thank the reviewer, for these concerns and suggestions, it is not for the research team to interfere in the delivery of routine services nor prescribe how routine HCWs are assigned duties or duty stations. We also hope that others will view that information with an eye for applying 2wT to those settings or investigating other more efficient pathways to provide follow-up care. Using 2wT would, indeed, reduce a lot of clinician time that appears wasted in transportation time in rural areas, especially. 

I am still troubled by the conclusion that the use of 2wT could yield savings of over $1.1 million per year (line 442). As stated above, the labor savings are not monetary savings so there is no budgetary reduction associated with these “savings”. Also, I’m not sure there is evidence to suggest men’s hesitancy to take up VMMC because of the need to return for follow-up visits and with more efficient follow-up that uptake would increase (lines 444-446).

Response: 

Transportation and associated costs for VMMC follow-up visits are a burden to men in SA, with much literature to suggest that many men try to avoid follow up visits (from migration, employment obligations, stigma, able to heal independently) and that there is overreporting of follow-up. We added 2 additional references [10, 11] to the introduction in support of that statement. Not requiring visits but providing this telehealth opportunity to interact, appears to be a win-win for men who receive support to heal well and their nurses who can provide this care without traveling. 

We rephrased the conclusion to remove the $1.1 million per year, to read, “Evidence from two RCTs in Zimbabwe and South Africa demonstrate that this 2wT approach provides high-quality VMMC follow-up as compared to required in-person reviews and lowers overall VMMC program costs. Rural savings using 2wT offset nominal increased costs in urban areas. Additional 2wT associated improvements in care quality, supervision, and verification also likely leading to longer-term savings. The health sector should invest in 2wT. In the context of National VMMC targets of 350,000 to 500,000 VMMCs per annum, employing 2wT could dramatically reduce the number of in-person post-operative reviews, resulting in concrete efficiency gains and significant annual program savings. Invested these resources back into the VMMC program could further expand VMMC access, improve care quality, and advance VMMC program goals of safe, efficient, and effective VMMC scale-up.”

Specific Comments:

Line 214 & 215 (& perhaps elsewhere) replace “Aes” with “AEs”. My guess is your word processing software was trying to be helpful .

Response: Completed. 

Line 242-244: The decision to only count the cost for a successful attempt even if multiple attempts were required seems to put downward bias on tracing costs. Please justify this decision.

Response: 

Documentation of attempts at tracing were poor. Therefore, we only have anecdotal support on previous attempts. This is the case and further supports the idea that 2wT could actually offset additional costs that were underestimated in this costing approach.

Line 360: According to Line 113, NDoH protocols already require two post-surgery follow-up visits. So why is this being treated as a hypothetical scenario? 

Response: 

Adherence to these guidelines varies widely, but the guidelines remain unchanged. With COVID19, there is also growing acceptance that telehealth or telephone follow-up may be acceptable. Although there are formal policies requiring 2 post-operative visits, we provided evidence that telehealth may be acceptable to replace in-person reviews for low risk men according to new PEPFAR documents about follow-up for low-risk men, but Low risk” has not been defined. The NDoH, PEPFAR, and the VMMC community need this evidence to further advocate for policy change – which is often slower than practice moves. 

---

## [Editor Report · Decision Letter 2]

2 Nov 2023

Cost savings in male circumcision post-operative care using two-way text-based follow-up  in rural and urban South Africa

PONE-D-23-01844R2

Dear Dr. Feldacker,

We’re pleased to inform you that your manuscript has been judged scientifically suitable for publication and will be formally accepted for publication once it meets all outstanding technical requirements.

Kind regards,

Lisa Suzanne Dulli, PhD

Academic Editor

PLOS ONE
---

## [Editor Report · Acceptance letter]

7 Nov 2023

PONE-D-23-01844R2 

Cost savings in male circumcision post-operative care using two-way text-based follow-up  in rural and urban South Africa 

Dear Dr. Feldacker:

I'm pleased to inform you that your manuscript has been deemed suitable for publication in PLOS ONE. Congratulations! Your manuscript is now with our production department. 

Kind regards, 

on behalf of

Dr. Lisa Suzanne Dulli 

Academic Editor

PLOS ONE